# Artificial Feeding of *Ornithodoros fonsecai* and *O. brasiliensis* (Acari: Argasidae) and Investigation of the Transstadial Perpetuation of *Anaplasma marginale*

**DOI:** 10.3390/microorganisms11071680

**Published:** 2023-06-28

**Authors:** Ana Carolina Castro-Santiago, Leidiane Lima-Duarte, Jaqueline Valeria Camargo, Beatriz Rocha De Almeida, Simone Michaela Simons, Luis Antonio Mathias, Ricardo Bassini-Silva, Rosangela Zacarias Machado, Marcos Rogério André, Darci Moraes Barros-Battesti

**Affiliations:** 1Department of Preventive Veterinary Medicine and Animal Health, School of Veterinary Medicine, University of São Paulo, São Paulo 05508-270, Brazil; ana_carolinacsantiago@hotmail.com (A.C.C.-S.); leidyliduarte123@gmail.com (L.L.-D.); 2Department of Pathology, Reproduction and One Health, Faculty of Agrarian and Veterinary Sciences, Paulista State University, Jaboticabal 14884-900, Brazil; jaque2911@hotmail.com (J.V.C.); beatriz.r.almeida@unesp.br (B.R.D.A.); la.mathias@unesp.br (L.A.M.); ricardo.bassini@gmail.com (R.B.-S.); rzacariasmachado@gmail.com (R.Z.M.); mr.andre@unesp.br (M.R.A.); 3Parasitology Laboratory, Butantan Institute, São Paulo 05503-900, Brazil; simone.simons@butantan.gov.br

**Keywords:** argasid ticks, biology, transmission, bovine anaplasmosis

## Abstract

*Anaplasma marginale* is a Gram-negative, obligate intraerythrocytic bacterium that causes bovine anaplasmosis. While hard ticks of the genera *Dermacentor* and *Rhipicephalus* can be biological vectors, transmitting this pathogen via saliva during blood meals, blood-sucking insects, and fomites play a role as mechanical vectors. Little is known about the interaction between *Anaplasma marginale* and Argasidae ticks. Among soft ticks, *Ornithodoros fonsecai* (Labruna and Venzal) and *Ornithodoros brasiliensis* Aragão inhabit environments surrounding localities where many cases of bovine anaplasmosis have been reported. Ticks of the species *O. fonsecai* parasitize bats, while *O. brasiliensis* can parasitize different vertebrate species. Therefore, the present study aimed to feed third-instar nymphs artificially (N3) of *O. fonsecai* and *O. brasiliensis* using blood samples obtained from a calf naturally infected with *A. marginale* and rabbit blood added to *A. marginale-*containing bovine erythrocytes, to investigate the ability of these nymphs to acquire, infect and transstadially perpetuate this agent. For the artificial feeding system, adapted chambers and parafilm membranes were used. Nymphs of both tick species were submitted to different replications weighed before and after each feeding. Blood samples and molted ticks were submitted to DNA extraction, quantitative real-time PCR for the *msp1*β gene to detect *A. marginale* DNA, while a semi-nested polymerase chain reaction for the *msp1*α gene was performed for genotyping. Using calf blood naturally infected with *A. marginale*, among the three artificial feeding replications performed with *O. fonsecai* and *O. brasiliensis* nymphs, the DNA of *A. marginale* was detected in both nymphs after 30–50 days of molting. For artificial feeding with rabbit blood added to bovine erythrocytes containing *A. marginale*, the DNA of this pathogen was also detected in both nymph species. As for the assay for the *msp*1α gene, strains were found Is9; 78 24-2; 25; 23; α; and β. It was concluded that nymphs (N3) of *O. fonsecai* and *O. brasiliensis* could feed artificially through a parafilm membrane using blood from calves and rabbits infected by *A. marginale*. The DNA of *A. marginale* was detected in nymphs fed artificially of both tick species studied after molt. However, further studies are needed to confirm transstadial perpetuation in other instars and their host transmission capacity.

## 1. Introduction

In the natural infection cycle of arthropod-borne agents, the sequence of events consists of ingesting a pathogen from an infected vertebrate host, its development in arthropod tissues, and its eventual transmission to a susceptible host animal [1]. “Transstadial transmission” and “transstadial infection” have been widely used to describe not only the passage but also the development of a pathogen from one tick stage to the next [1]. Biologically, *Anaplasma marginale* (Rickettsiales: Anaplasmataceae) can be transmitted by approximately 20 hard tick species belonging mainly to the genera *Dermacentor*, *Rhipicephalus*, and *Hyalomma* [2]. In tropical and subtropical regions, the species *Rhipicephalus annulatus* (Say) is the most reported. In cold and temperate regions, the species *Dermacentor andersoni* Stiles, *Dermacentor variabilis* (Say), and *Dermacentor albipictus* (Packard), and distributed worldwide, the species *Rhipicephalus microplus* (Canestrini) is reported [3,4,5,6]. Other tick species, such as *Rhipicephalus sanguineus* s.l. (Latreille) and *Rhipicephalus simus* (Koch) have also been incriminated in the transstadial transmission of this bacterium [3]. Despite little knowledge of the relationship between Anaplasmataceae agents and Argasidae ticks, studies have shown evidence of *Anaplasma* spp. DNA in soft ticks, such as *Argas vespertilionis* (Latreille) in France [7], *Argas persicus* (Oken) in Algeria [8], *Ornithodoros spheniscus* (Hoogstraal, Wassef, Hays & Keirans) in Chile [9] and *Ornithodoros hasei* in Brazil [10].

*Anaplasma marginale* is an obligate intraerythrocytic Gram-negative α-proteobacterium. It is the causative agent of bovine anaplasmosis, associated with significant economic losses for livestock industries. Recently, this agent has also been detected in other animal species [3,11,12,13]. Transmission of this agent occurs mechanically through the bites of flies or fomites contaminated with blood or biologically by ticks [14,15,16,17]. This agent can be transmitted from stage to stage (interstage or transstadial) or intrastadial, which affects male ticks which interrupt feeding on an infected host and continue the blood meal on a susceptible host, thus transmitting *A. marginale*. Transovarian transmission has not been reported [3,18].

Different techniques of artificial feeding (or in vitro feeding) have been used since the first published studies [19,20] and applied in the laboratory to a wide range of hematophagous arthropods, with the aims of studying tick biology, vector-pathogen interactions, and colony maintenance. Techniques that use natural or artificial membranes offer relative simplicity and have the potential to reduce or avoid the use of animals as hosts in colony maintenance [21,22].

Considering that the soft ticks *Ornithodoros fonsecai* (Labruna and Venzal) and *Ornithodoros brasiliensis* Aragão can be found in environments close to locations with reports of bovine anaplasmosis and are a parasitic species of bat and different species of vertebrates, respectively, the objective of the present study was to feed artificially created third instar (N3) nymphs of *O. fonsecai* and *O. brasiliensis* using blood from a calf naturally infected with *A. marginale*, and rabbit blood added to *A. marginale-*containing bovine erythrocytes, to investigate the ability of these nymphs to acquire, to become infected with, and transstadially perpetuate this agent. 

## 2. Material and Methods

### 2.1. Origin of Ticks and Maintenance of Colonies

Specimens of *O. fonsecai* were collected in caves located in the municipality of Bonito (21°06′31″ S, 56°34′44″ W), state of Mato Grosso do Sul, central-western Brazil, while specimens of *O. brasiliensis* were collected from the ground in the municipality of São Francisco de Paula (29°20′00″ S; 48°30′21″ W), state of Rio Grande do Sul, southern Brazil. The ticks were kept in biological oxygen demand (BOD) incubators (Eletrolab, São Paulo, Brazil) at 25 ± 1 °C (*O. fonsecai*) and 27 ± 1 °C (*O. brasiliensis*) and at a relative humidity of 90 ± 10%. Both tick colonies were maintained at the Laboratory of Immunoparasitology, Department of Pathology, Reproduction and One Health, São Paulo State University (UNESP), Jaboticabal, São Paulo, Brazil, and at the Laboratory of Parasitology, Butantan Institute (#5740291018). 

To maintain both tick species and obtain nymphs for artificial feeding from all generations used in the present study, adults and larvae were fed to New Zealand rabbits (*Oryctolagus cuniculus*). The tick feeding was authorized by the Animal Use Ethics Committee (#9727130718) of the Faculty of Veterinary Medicine and Zootechnics, University of São Paulo, São Paulo, Brazil.

### 2.2. Artificial Feeding of Nymphs

#### 2.2.1. Acquisition of *Anaplasma marginale*-Infected Calf Blood Samples

Three artificial feeding replications using blood from a calf naturally infected with *A. marginale* were performed to feed nymphs of both tick species. These experiments were authorized by the Animal Use Ethics Committee (#01952/18) of UNESP, Jaboticabal, São Paulo.

In each replication, blood samples were collected using heparin-containing tubes from randomly selected calves (*Bos taurus taurus*) that were kept at the Department of Cattle-rearing of the same institution. An aliquot of each sample was separated for DNA extraction and molecular analysis. DNA extraction was performed using the DNeasy Blood and Tissue kit (Qiagen^®^, Hilden, Germany), following the manufacturer’s recommendations. To check for the presence of inhibitors in the DNA samples, a conventional polymerase chain reaction (cPCR) protocol for the endogenous glyceraldehyde-3-phosphate dehydrogenase (*gapdh*) gene was performed, as shown in Appendix A [23]. The PCR assays used are described in Appendix A. The cPCR products were separated using electrophoresis on 1% agarose gel stained with ethidium bromide (Life Technologies^®^, Carlsbad, CA, USA). They were viewed through UV light using the Image Lab software, version 4.1 (Bio-Rad^®^, Hercules, CA, USA).

DNA samples were subjected to a qPCR reaction for *A. marginale* based on the *msp1*β gene, as shown in Appendix A [24]. Quantification was performed using plasmid pSMART (Integrated DNA Technologies^®^, Coralville, IA, USA) containing a fragment of the *A. marginale msp1*β gene. Serial dilutions were performed to construct standards with different concentrations of plasmid DNA containing the target sequence (2.0 × 10^7^ copies/μL to 2.0 × 10^1^ copies/μL). Plasmid copy number was determined from the following formula: (Xg/μL DNA/[plasmid size (bp) × 660]) × 6.022 × 10^23^ × plasmid copies/μL). Ultrapure sterile water (Qiagen^®^, Madison, WI, USA) and an aliquot of DNA from the Jaboticabal strain of *A. marginale* were used as negative and positive controls in the qPCR assays, respectively. The remaining blood samples were used to feed the ticks. 

Three feeding repetitions were carried out for both species containing 15 third instar nymphs (N3). The nymphs were weighed as pools before and after feeding. For feeding, containment chambers were constructed consisting of 50 mL Falcon tubes cut, coated with parafilm, and sealed with adhesive tape coupled to a 6-well plate. Before the chambers were sealed, the parafilm was rubbed onto the calves’ skin. Thus, after sealing, calf hair was seen to be spread over the parafilm. Approximately 3 mL of blood was dispensed into each well. The feeding system consisted of a bench shaker with a magnetic agitator and heating. 

After the molting period, the nymphs were subjected to DNA extraction and conventional PCR (cPCR) assay based on the endogenous mitochondrial 16S rRNA gene, as shown in Appendix A [25]. For this purpose, DNA was individually extracted from the tick specimens using the DNeasy Blood and Tissue kit (Qiagen^®^, Hilden, Germany), following the manufacturer’s recommendations. The cPCR products were separated utilizing electrophoresis on 1% agarose gel stained with ethidium bromide (Life Technologies^®^, Carlsbad, CA, USA). They were viewed through UV light using Image Lab software, version 4.1 (Bio-Rad^®^, Hercules, CA, USA). The nymphs’ DNA samples were subjected to qPCR for *A. marginale*, based on the *msp1*β gene.

#### 2.2.2. Rabbit Blood + *A. marginale*-Infected Bovine Erythrocytes

For artificial feeding using rabbit blood added to *A. marginale-*containing bovine erythrocytes, four replications of these collections and feedings were performed. Blood collected from rabbits was immediately transferred to a heparin-containing collection tube. Cryovials of 1.5 mL containing the Jaboticabal strain of *A. marginale-*infected bovine erythrocytes kept in liquid nitrogen were manually thawed. Then, *A. marginale-*infected bovine erythrocytes were washed twice with Leibovitz L-15 medium (Vitrocell^®^, Campinas, SP, Brazil) at a 1:1 ratio under centrifugation at 4000× *g* for 15 min at 15 °C each. 

The pellet thus formed (approximately 5 mL) was added into tubes containing rabbit blood samples in each replication. An aliquot of the blood infected with *A. marginale* was separated for DNA extraction and molecular analysis. The DNA extraction, cPCR for the endogenous *gapdh* gene, and qPCR for *A. marginale* based on the *msp1*β gene were performed as previously described. For the three repetitions of the artificial feeding experiments for nymphs of both tick species, 15 third instar (N3) nymphs were used for each one. Fifteen N3 nymphs were separated into the control groups of each tick species for feeding on hosts (rabbits). 

All nymphs in both the control group and the replications were weighed in pools before and after feeding. For feeding, the same system as described above was used. Approximately 30–50 days after the molting period, ticks were subjected to DNA extraction, cPCR for the endogenous mitochondrial 16S rRNA gene, and qPCR for *A. marginale* based on the *msp1*β gene.

#### 2.2.3. Semi-Nested PCR for *A. marginale* Based on the *msp1*α Gene and Confirmation of Transstadial Perpetuation

All the tick samples that were positive for the qPCR assay for *A. marginale* based on the *msp1*β gene from all experiments were subjected to a semi-nested PCR (snPCR) based on the *msp1*α gene, as shown in Appendix A [26]. The snPCR products were separated using electrophoresis on 1% agarose gel stained with ethidium bromide (Life Technologies^®^, Carlsbad, CA, USA). They were viewed through UV light using the Image Lab software, version 4.1 (Bio-Rad^®^, Hercules, CA, USA). Subsequently, to confirm the absence of remnant blood from the host (calf or rabbit), qPCR-positive samples were subjected to cPCR targeting the mammal *gapdh* gene, as described above.

#### 2.2.4. Purification, Sequencing of Amplified Products, and Analysis of Consensus Sequences Based on the *msp1*α Gene

The products based on the *msp1*α gene obtained through snPCR were purified using the ExoSAP-IT™ PCR Product Cleanup Reagent kit (Thermo Scientific^®^, Waltham, MA, USA), following the manufacturer’s recommendations. Sequencing of the amplified products was performed using an automated technique based on the dideoxynucleotide chain termination method [27] in the ABI PRISM 3730 DNA Analyzer sequencer (Applied Biosystem^®^, Waltham, MA, USA). The sequencing reactions were performed using the BigDye™ Terminator v3.1 cycle sequencing kit, and sequencing was conducted at the Human Genome and Stem Cell Research Center located at the Biosciences Institute of the University of São Paulo, state of São Paulo, Brazil. Electropherogram results were analyzed using the Phred-Phrap software, version 23 [28]. The quality of each nucleotide sequence was checked for a score and was considered to present good quality when scoring Phred > 20. Consensus sequences obtained through the alignment of the forward and reverse sequences were constructed using the same software. Electropherograms that showed double peaks were not included in subsequent analyses and were considered possible co-infections. The sequences generated were input to the BLASTn software (version 2.7.1) [29] to determine the closest similarities to the *msp1*α gene sequences previously deposited in GenBank [30].

#### 2.2.5. Classification of *A. marginale* Genotypes

To identify the genotype and the classification in accordance with Estrada-Peña et al. [31], we followed these steps: the region called Shine-Dalgarno (GTAGG) was found, followed by the next translation initiation codon (ATG). The repeated sequence analysis was performed in accordance with the nomenclature proposed [32]. The SD-ATG distance was calculated using the formula (4 × m) + (2 × n) + 1, as previously described [31]. The consensus sequences thus obtained were transformed into amino acids using the ExPaSy translation tool [http://web.expasy.org/translate/ (accessed on 1 November 2020)] from the Swiss Institute of Bioinformatics, and their amino acid variability was analyzed using the *RepeatAnalyzer* software (version 3.0) [33].

## 3. Statistical Analysis

The effect of the variables “Blood” and “Species” (explanatory) on the variables “Fed” and “Positive” (outcomes) was investigated. Univariate analysis was performed using Fisher’s exact test. Logistic regression analysis was also performed to obtain models with explanatory variables that had a significant effect (*p* < 0.05). For this analysis, R was used, considering a significance level of 5% (α = 0.05). In the univariate analysis, the proportion of units with the outcome “Fed” among the “Exposed” nymphs was analyzed. In the outcome “Positive”, the proportion between “Exposed” nymphs, between “Fed” nymphs, and between “Molt” nymphs were analyzed.

## 4. Results

The total number of N3 nymphs of both tick species used in the artificial feeding was 180 specimens, while 30 N3 nymphs were used as control. 

### 4.1. Nymphs Artificially Fed on Blood from Calves Naturally Infected with A. marginale

From the three blood collections and DNA extraction, the aliquots of calf blood were found to be positive in the cPCR assay for the *gapdh* gene and the qPCR assay for the *msp1*β gene, with quantifications of 10^5^, 10^5,^ and 10^6^ copies of the *msp1*β/μL DNA fragment in the first, second and third replications, respectively. Considering the total number of N3 nymphs of *O. fonsecai*, 34 specimens fed on blood from calves and, out of these, 14 (41.1%) molted to 12 N4, 1 female and 1 male between 34–80 days. Twenty N3 died after feeding (58.8%), as shown in Table 1. After molting of nymphs and the qPCR assay for the *msp1*β gene of *A. marginale*, six specimens (four nymphs and two adults) were found to be negative when the assay was performed after 48 days, while eight N4 nymphs were positive after 34 days, with quantification between 10^3^–10^4^ copies of the *msp1*β/μL DNA fragment, as shown in Table 1. These eight *A. marginale*-positive samples were negative in the cPCR for the mammal *gapdh* gene. Regarding *O. brasiliensis*, 31 N3 nymphs were fed, and of these, 22 molted to N4 between 30–57 days after feeding, as shown in Table 1. After molting for 31 days, 9 N4 nymphs were negative for the *msp1*β gene, and 13 N4 nymphs were positive with quantification between 10^1^–10^2^ copies of the *msp1*β/μL DNA fragment, as shown in Table 1. All positive samples were negative in the cPCR for the mammal *gapdh* gene. The cPCR for the mitochondrial gene 16S rRNA performed after the molting of the nymphs was positive for each replication.

### 4.2. Nymphs Artificially Fed on Rabbit Blood Added to A. marginale-Infected Bovine Erythrocytes

Through the four blood collections and DNA extraction, the aliquots of rabbit + bovine blood samples were found to be positive in cPCR for the *gapdh* gene and in qPCR for the *msp1*β gene, with quantification of 10^6^ copies of the *msp1*β/μL DNA fragment. A total of 22 N3 nymphs of *O. fonsecai* were fed, and of these, 10 specimens molted to 9 N4 and 1 female between 35–64 days, as shown in Table 1. The qPCR assay was performed 37 days after molting, resulting in 2 N4 and the female positive for the *msp1*β gene, with quantification of 10^1^ copies of the *msp1*β/μL DNA fragment, while the 7 N4 nymphs were negative after 43 days after molting, as shown in Table 1. The remaining N3 nymphs that fed died. Regarding N3 nymphs of *O. brasiliensis*, 33 N3 were fed, and all molted to 25 N4 and one female between 28–35 days, as shown in Table 1. A total of 19 nymphs were negative in a qPCR assay for *A. marginale* 33 and 40 days after molting. In comparison, 13 nymphs and the female were positive after 33 days, with the quantification between 10^2^–10^3^ copies of the *msp1*β/μL DNA fragment, as shown in Table 1. All *A. marginale-*positive samples from both species were negative in the cPCR targeting the mammal *gapdh* gene. The cPCR for the mitochondrial gene 16S rRNA, performed after the molting of nymphs, was positive for each replication.

All 15 N3 nymphs of the control group of *O. fonsecai* fed on hosts (rabbits) and molted to N4 between 32–38 days, with no deaths. Likewise, 93.3% of the N3 nymphs of *O. brasiliensis* fed on hosts (rabbits), and all molted to N4 in 29 days, with no death records (Table 1).

The BLASTn analysis on the *msp1*α gene of the *A. marginale* sequences obtained from artificially fed *O. fonsecai* and *O. brasiliensis* nymphs is shown in Table 2. Although 10 samples showed amplification in the qPCR (*msp1*β gene) and snPCR (*msp1*α gene), only 5 samples showed good sequencing qualities. Thus, only these were used for subsequent analyses. Through BLASTn, they presented similarities ranging from 96.4% to 100% to sequences of *A. marginale* (KJ626203) and (CP023731) that had previously been deposited in GenBank, with a coverage of 100%. The classifications of *A. marginale* genotypes and strains of positive samples for the *msp1*α gene are shown in Table 3; six strains were found, Is9; 78 24-2; 25; 23; α; and β.

Considering the general data regarding the different blood types, the chance of engorged nymphs being fed calf blood was significantly higher than those fed rabbit blood added to infected bovine erythrocytes (*p* = 0.0133). When analyzed simultaneously, the species did not significantly affect the frequency of fed nymphs. Still, when analyzed separately, nymphs of *O. fonsecai* had a significant difference when fed with calf blood compared to rabbit blood added to erythrocytes (*p* = 0.0162). On the other hand, this difference was not significant for *O. brasiliensis* nymphs (*p* = 0.3818).

Based on the univariate analysis of the frequency of positive nymphs for the *msp1*β gene of *A. marginale*, when the variable “Blood” was used, the general data showed that the chance of occurrence of positive nymphs when fed with calf and rabbit blood added to erythrocytes did not differ significantly (*p* = 0.5841). When the variable “Species” was used, the general data showed significance (*p* = 0.005679). The probability of *O. fonsecai* nymphs being positive was significantly lower than the chance of this in *O. brasiliensis* nymphs. And when analyzed separately, the species *O. fonsecai* and *O. brasiliensis* were insignificant, obtaining the values *p* = 0.1966 and *p* = 1.00, respectively.

## 5. Discussion

In the present study, we demonstrated the artificial feeding of *O. fonsecai* and *O. brasiliensis* nymphs using a parafilm membrane. In addition, we showed the ability of these nymphs to acquire and transstadially perpetuate *A. marginale* after feeding on calf blood naturally infected with *A. marginale* and on rabbit blood added to infected bovine erythrocytes.

For *O. fonsecai* nymphs, the number of specimens engorged with infected calf blood (75.5%) was greater than those fed rabbit blood (48.8%); therefore, the average weight gain of nymphs fed with calf blood was bigger. As for the molting rate, those fed with rabbit blood changed more (45.4%) than those fed with calf blood (41.1%). However, more nymphs died when fed calf blood (58.8%), while 54.5% died when fed with rabbit blood plus bovine erythrocytes containing *A. marginale*.

As for *O. brasiliensis* nymphs, the feeding rate (73.3%) with rabbit blood added with bovine erythrocytes containing *A. marginale* was higher when compared with those engorged using calf blood (68.8%). The same occurred regarding the molting rate, being 100% and 70.9% for the respective types of diet. However, more nymphs died when fed calf blood (29%), while no deaths were recorded when fed rabbit blood plus bovine erythrocytes containing *A. marginale*.

In all replications of artificial feeding for both species of ticks, the parafilm used as the membrane was rubbed on the skin of calves and rabbits. Additionally, calf and rabbit hairs were put into the containment chambers to stimulate tick feeding. Interestingly, this type of membrane allowed nymphs from both *Ornithodoros* species to be easily manipulated, such that there was no blood leakage by the time the specimens were attached and engorged.

In a previous study, 90% of *O. fonsecai* N2 nymphs artificially fed on rabbit blood through parafilm membrane were able to feed and molt, with no record of deaths. In addition, blood was spread over the blood bag as a stimulant to attract the nymphs [36]. In another study, when unfed *Ornithodoros coriaceus* Koch nymphs and adults were allowed to feed on a parafilm membrane surface with or without a thin layer of guinea pig hair, the presence of this stimulant was associated with a higher rate of engorgement [37].

In a comparative study evaluating the feeding behavior of *Ornithodoros tholozani* (Laboulbène and Mégnin) and *Ornithodoros moubata* (Murray) using a silicone membrane and a modified Baudruche membrane and using rabbit ear wax as a host stimulus, a positive correlation between the feeding rate and the attachment speed was observed for both two tick species [38]. When using silicone membrane and defibrinated rabbit blood, 85% of the *O. moubata* nymphs and 92% of the *O. tholozani* nymphs became engorged. On the other hand, 85% of the *O. moubata* nymphs and 100% of the *O. tholozani* nymphs became engorged when subjected to the same diet and stimulus when using the modified Baudruche membrane [38].

Artificial feeding of *Ornithodoros tartakovsky* Olenev adults using parafilm as a membrane and defibrinated rabbit or horse blood generated successful engorgement of 89% (172/193) of the ticks (92% of the females and 85% of the males became engorged) [39]. These authors also limited the blood volume, so they needed approximately 2 to 9 mL to feed the ticks. In the present study, the volume used for feeding both species was approximately 3 mL, which was necessary for the total engorgement of the nymphs. In another study, when *Ornithodoros turicata* (Dugès) N5 nymphs and adults were artificially fed on parafilm membrane and defibrinated calf blood, all the specimens fed; unfortunately, the nymph molt rate and deaths among the tick specimens were not measured [40].

In a previous study, *O. moubata* nymphs and adults were subjected to artificial feeding to test several types of diets and anticoagulants [22]. For that purpose, heparinized bovine blood, heparinized and hemolyzed bovine blood, heparinized rat blood, and defibrillated sheep blood were used, with parafilm as the membrane through which artificial feeding took place. Preference for the first type of diet was demonstrated: this was the most satisfactory among the options, such that the mortality rate among fed females was less than 5%.

More recently, Alsever (a solution prepared with glucose, sodium citrate, and sodium chloride as an anticoagulant) was shown to be efficient for the artificial feeding of *O. fonsecai* nymphs [36] under laboratory conditions. In the present study, heparin was the anticoagulant used in artificially feeding *O. fonsecai* and *O. brasiliensis* nymphs in all replicates. This demonstrated that even with a significant mortality rate, it was possible to obtain nymphs that molt to the adult or next nymphal stage.

In a previous study, artificial feeding of *D. andersoni* nymphs and adults with bovine red blood cells derived from heparinized bovine blood infected with *A. marginale* and using rabbit and mouse skin as the stimulant surface, both nymphs and adults were able to acquire the bacteria. Indeed, nymphs perpetuated the pathogen and were able to infect naïve animals [41]. Moreover, when the residual blood containing tick saliva was collected from the artificial feeding system after tick engorgement and was inoculated into susceptible calves, the calves did not acquire the pathogen.

Up to now, studies in which artificial feeding of ticks with *A. marginale* were done had the aim of analyzing the acquisition of the pathogen by investigating the presence of the pathogen in the salivary glands and midgut of infected hard ticks and also through ascertaining their capacity to transmit the bacterium to a susceptible host. A study using capillary tubes to artificially feed *D. variabilis* adults with *A. marginale*-infected blood demonstrated the presence of bacterial DNA in the tick midgut, albeit at low levels; on the other hand, *A. marginale* DNA was not detected in the tick salivary glands. When subjected to feeding on the host, a sufficiently detectable level of rickettsemia was observed in the salivary glands [42].

Recently, male *D. andersoni* ticks were subjected to artificial feeding on calf blood through a silicone membrane administering four doses with different concentrations of *A. marginale* to investigate the acquisition of the bacteria. After complete digestion of the blood, some specimens were dissected, and the salivary glands and midgut were collected. The remaining males were artificially fed again to ascertain whether the pathogen was transmitted in uninfected blood. It was demonstrated that the ticks were able to acquire *A. marginale* through artificial feeding, with a midgut infection rate ranging from 80 to 100% when subjected to feeding at higher concentrations of the bacteria; on the other hand, the infection rate in the salivary glands was 72%, among ticks subjected to the same bacterial concentrations [43].

In the present study, after the molting of the nymphs fed on calf blood, 42.8% of the *O. fonsecai* specimens were found to be negative in the qPCR assay for the *msp1*β gene of *A. marginale* 48 days after ecdysis. However, 57.1% of the molted nymphs were positive after 34 days. On the other hand, when nymphs were fed on rabbit blood plus bovine erythrocytes, 2 N4 and the female of *O. fonsecai* (30%) were positive for *A. marginale* when the qPCR assay was performed 37 days after the molt, and the remaining nymphs (70%) were negative 43 days after molting.

Of the *O. brasiliensis* specimens, 40.9% of nymphs fed on calf blood were negative for qPCR for the *msp1*β gene after 31 days of molting, while 59% were positive. Those fed with rabbit blood plus bovine erythrocytes, 57.5% were negative for *A. marginale* 40 days after molting, while 42.4% were positive between 33–40 days. The reason for this variation in qPCR positivity among the days after the molt remains unclear. Previously, a laboratory experiment was conducted to investigate transstadial perpetuation and transovarian transmission of *Borrelia coriaceae* among nymphs and adults of *O. coriaceus.* It was found after feeding the nymphs and performing tissue dissection between 15 and 60 days after the molt that spirochetes were only present in 7% of the molted nymphs [44]. These authors suggested that the transstadial perpetuation of *B. coriaceae* in *O. coriaceus* might not be 100% efficient and that some nymphs might lose their infection during or after ecdysis. Regarding transovarian transmission, 14% of the larvae were infected.

Based on the genotyping of the *A. marginale msp1*α gene, it was noted that the *A. marginale* strains found in tick specimens evaluated in the present study had low diversity. Using the genotype classification suggested by Estrada-Peña et al. [31], two different genotypes were observed, namely genotypes H and F. To our knowledge, this was the first study performed on argasid ticks to investigate the genetic diversity of the *A. marginale* strain acquired by artificially fed ticks.

None of the nymphs of both species studied here retained residual blood after the molt when analyzed for the *gapdh* gene. Thus, suggesting that transstadial perpetuation of *A. marginale* was occurring between the N3 and N4 instars of these ticks.

## 6. Conclusions

The artificial feeding system using parafilm membrane and heparin as an anticoagulant, it was possible to feed *O. fonsecai* and *O. brasiliensis* N3 nymphs on blood from calves naturally infected with *A. marginale* and on rabbit blood added to *A. marginale*-containing bovine erythrocytes.

According to the results shown, the nymphs of *O. fonsecai* fed better on calf blood, but the molting and mortality rates were better when fed on rabbit blood plus bovine erythrocytes. Regarding the number of engorgements, the *O. brasiliensis* nymphs were fed equally with both types of diet; however, the molt and death rates were better when they ingested rabbit blood.

The role of *Ornithodoros* spp. in the transmission of *A. marginale* between cattle is still uncertain. However, it cannot be ruled out since the present study verified the possible transstadial perpetuation of the bacteria, which already shows that this type of pathogen maintenance can occur in argasid ticks.

Therefore, the possibility of transstadial perpetuation of this pathogen in both argasid tick species studied here should not be disregarded. Further studies to investigate the ability of *Ornithodoros* nymphs to transmit *A. marginale* to naive cattle are needed.

## Figures and Tables

**Table 1 microorganisms-11-01680-t001:** Parameters related to artificial feeding of the species *O. fonsecai* and *O. brasiliensis*: experimental group, blood type, replication, number of nymphs exposed and fed, number of molts, if positive or negative for the qPCR assay for the *msp1*β gene for *A. marginale*, time after molt (in days) and quantification of the bacterial copy number.

Blood	Replication	Species	N		Molt	N		Time after Molt (Days)	Quantification
			Exposed	Fed	N	Positive	Negative		
Calf	1st	*O. fonsecai*	15 N3	9 N3	4 N4, 1F, 1M	-	4 N4, 1F, 1M	48	-
		*O. brasiliensis*	15 N3	3 N3	-	-	-	-	-
	2nd	*O. fonsecai*	15 N3	13 N3	8 N4	8 N4	-	34	10^3^–10^4^
		*O. brasiliensis*	15 N3	15 N3	12 N4	7 N4	5 N4	31	10^1^
	3rd	*O. fonsecai*	15 N3	12 N3	-	-	-	-	-
		*O. brasiliensis*	15 N3	13 N3	10 N4	6 N4	4 N4	31	10^2^
Rabbit	1st	*O. fonsecai*	15 N3	7 N3	7 N4	-	7 N4	43	-
		*O. brasiliensis*	15 N3	9 N3	9 N4	3 N4	6 N4	40	10^2^
	2nd	*O. fonsecai*	15 N3	9 N3	-	-	-	-	-
		*O. brasiliensis*	15 N3	10 N3	10 N4	4 N4	6 N4	40	10^2^
	3rd	*O. fonsecai*	15 N3	6 N3	2 N4, 1F	2 N4, 1F	-	37	10^1^
		*O. brasiliensis*	15 N3	7 N3	6 N4, 1F	6 N4, 1F	-	33	10^2^–10^3^
Control		*O. fonsecai*	15 N3	15 N3	15 N4	-	-	-	-
		*O. brasiliensis*	15 N3	14 N3	14 N4	-	-	-	-

**Table 2 microorganisms-11-01680-t002:** BLASTn analysis on *msp1*α gene of *A. marginale* sequences obtained from artificially fed *O. fonsecai* and *O. brasiliensis* nymphs.

				BLASTn			
Blood	Replication	Species	Number of Nymphs	Similarity	Coverage	Sequence	References
Calf	1st	*O. fonsecai*	2 N4	96–96.9%	100%	*Anaplasma marginale* (KJ575590) and (KJ575560)	[34]
Rabbit	3rd	*O. brasiliensis*	3 N4	96.4–100%	100%	*Anaplasma marginale* (KJ626203) and (CP023731)	[35]

**Table 3 microorganisms-11-01680-t003:** Classification of *A. marginale msp1*α genotypes and strains obtained from artificially infected *O. fonsecai* and *O. brasiliensis* nymphs, as assessed through *RepeatAnalyzer*, according to experiment replication number, tick species, instar, and estimated absolute rickettsemia.

Blood	Replication	Species	Sample	Absolute Rickettsemia (*msp1*β/μL)	Genotype	Strain
Calf	1st	*O. fonsecai*	1 N4	2.79 × 10^3^	*	Is9, 78 24-2 25
			1 N4	6.9 × 10^4^	F	Is9, 78 24-2 25
Rabbit	3rd	*O. brasiliensis*	1 N4	1.42 × 10^2^	H	23
			1 N4	6.17 × 10^2^	H	α β

*: unidentified genotype. The underline was used to separate the different strains.

## Data Availability

All data presented in this study are available in the article.

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
