# Peer review of "Artificial Feeding of Ornithodoros fonsecai and O. brasiliensis (Acari: Argasidae) and Investigation of the Transstadial Perpetuation of Anaplasma marginale"

_microorganisms, 2023, doi:10.3390/microorganisms11071680_

Round 1
Reviewer 1 Report
The manuscript addresses a topic relevant to the health area and brings relevant data to be published. The writing is dense and long. If possible, I suggest optimizing it. I send small points of suggestion to be observed by the authors:
Introduction- Lines 65-66: Would not be repeating the information already written in lines 49-50? Review.
Introduction- In this section of the manuscript, no information is provided about Ornithodoros fonsecai and Ornithodoros brasiliensis that would justify carrying out the study. This information is present only in the Abstract (lines 20-23). I suggest including this data in the Introduction as well.
Material and methods- Table 1: I suggest that Table 1 be included as supplemental material.
Material and methods- In general, the methodology is very extensive. I suggest, if there is no description of any new methodology, just cite the reference of the methodology used.
None.
Author Response
Response to Reviewer 1 Comments
Dear Reviewer 1,
Thanks for the suggestions made. Here are the point-by-point response:
Introduction- Lines 65-66: Would not be repeating the information already written in lines 49-50? Review.
Response: OK, reviewed
Introduction- In this section of the manuscript, no information is provided about Ornithodoros fonsecai and Ornithodoros brasiliensis that would justify carrying out the study. This information is present only in the Abstract (lines 20-23). I suggest including this data in the Introduction as well.
Reply: OK, reviewed
Material and methods- Table 1: I suggest that Table 1 be included as supplemental material.
Reply: OK
Material and methods- In general, the methodology is very extensive. I suggest, if there is no description of any new methodology, just cite the reference of the methodology used.
Reply: The methodology was developed during the experiment and there is no possibility to optimize it
Reviewer 2 Report
The contents of this manuscript is interest and the paper is scientifically well prepared and conducted. I have few inquiries to the authors:
1. why did they didn't do PCR on the collected ticks to be insure free from A. marginale?
2. Field significance of this work
3. The authors supposed that these soft tick could be infected and also a vector for A. marginale, but feeding of the infected ticks on calves free to show infection transmission is possible or not, didn't occur, what about of the dose of incolum?
Author Response
Response to Reviewer 2 Comments
Dear Reviewer 2,
Thanks for the suggestions made. Here are the point-by-point response:
1. why did they didn't do PCR on the collected ticks to be insure free from A. marginale?
Response: PCR was performed on 5 individuals of each species. Data were not demonstrated, but the result was negative for A. marginale for both species.
2. Field significance of this work
Response: The present work demonstrated that the pathogen, once detected in the post-molt nymphs, infers the possibility that the elucidated species may pose a risk to the areas where they are found.
3. The authors supposed that these soft tick could be infected and also a vector for A. marginale, but feeding of the infected ticks on calves free to show infection transmission is possible or not, didn't occur, what about of the dose of incolum?
Response: We are currently maintaining the colony to demonstrate the potential for transmission of infected nymphs on free calves.
Reviewer 3 Report
Review of the article by:
Ana Carolina Castro-Santiago, Leidiane Lima-Duarte, Jaqueline Valeria Camargo, Beatriz Rocha De Almeida, Simone Michaela Simons, Luis Antonio Mathias, Ricardo Bassini-Silva, Rosangela Zacarias Machado, Marcos Rogério André and Darci Moraes Barros-Battesti
Title:
Artificial feeding of Ornithodoros spp. (Acari: Argasidae) and investigation of the transstadial perpetuation of Anaplasma marginale
The submitted paper contains important and interesting data on the relativeness and interactions between Anaplasma marginale and two species of Argasidae ticks. Ticks of the species Ornithodoros fonsecai parasitize bats while Ornithodoros brasiliensis is able to parasitize different vertebrate species. The aim of the study was to artificially feed third-instar nymphs of O. fonsecai and O. brasiliensis using blood samples obtained from a calf naturally infected with Anaplasma marginale, and rabbit blood added to A. marginale-containing bovine erythrocytes, in order to investigate the ability of these nymphs to acquire, infect and transstadially perpetuate this agent. The authors concluded that nymphs of O. fonsecai and O. brasiliensis were able to feed artificially through a parafilm membrane using blood from calves and rabbits infected by A. marginale. Moreover, the DNA of A. marginale was detected in nymphs fed artificially of both tick species studied, after molt.
The text of the paper is well-defined, materials and methods are described sufficiently, results are presented in adequate detail, and the discussion is appropriate. Tables are informative and references comprehensive.
Overall, considering the content of the paper, I am of an opinion that the article fits well into the scope of Microorganisms and should certainly be published without revision and correction.
Few minor suggestions are marked in the text enclosed.
Conclusion: publish without revision.
Grzegorz Gabrys Zielona Gora, 5 June 2023

Author Response
Response to Reviewer 3 Comments
Dear Reviewer 3,
Thanks for the suggestions made. Here are the point-by-point response:
Few minor suggestions are marked in the text enclosed.
Response: All minor suggestions made in the text were applied.